# COVID-19 Pandemic and Agroecosystem Resilience: Early Insights for Building Better Futures

**Lalisa A. Duguma** [1,*], **Meine van Noordwijk** [1,2], **Peter A. Minang** [1] and **Kennedy Muthee** [1]

1   World Agroforestry (ICRAF), UN Avenue, Gigiri, P.O. Box 30677, Nairobi 00100, Kenya;
    M.VANNOORDWIJK@CGIAR.ORG (M.v.N.); A.MINANG@CGIAR.ORG (P.A.M.);
    K.MUTHEE@CGIAR.ORG (K.M.)
2   Plant Production Systems, Wageningen University, 6708 PB Wageningen, The Netherlands
*   Correspondence: L.A.DUGUMA@CGIAR.ORG

**Abstract:** The way the COVID-19 pandemic disrupted human lives and livelihoods constituted a stress test for agroecosystems in developing countries, as part of rural–urban systems and the global economy. We applied two conceptual schemes to dissect the evidence in peer-reviewed literature so far, as a basis for better understanding and enabling 'building back better'. Reported positive impacts of the lockdown 'anthropause' on environmental conditions were likely only short-term, while progress towards sustainable development goals was more consistently set back especially for social aspects such as livelihood, employment, and income. The loss of interconnectedness, driving loss of assets, followed a 'collapse' cascade that included urban-to-rural migration due to loss of urban jobs, and illegal exploitation of forests and wildlife. Agricultural activities geared to international trade were generally disrupted, while more local markets flourished. Improved understanding of these pathways is needed for synergy between the emerging adaptive, mitigative, transformative, and reimaginative responses. Dominant efficiency-seeking strategies that increase fragility will have to be re-evaluated to be better prepared for further pandemics, that current Human–Nature interactions are likely to trigger.

**Keywords:** anthropause; COVID-19; pandemic; impact pathways; natural resources; developing countries; resilience; restoration





## 1. Introduction

A pandemic like COVID-19 had been predicted [1,2], as the pool of potential zoonotic diseases (beyond MERS, SARS, and Ebola) remains large and is by no means restricted to bats or primates [3]. The cascading responses across virtually all sectors of societies to the SARS-CoV-2 virus causing the COVID-19 pandemic took the world by surprise in 2020 [4]. Within a time-span of a few months, public debate progressed from (failed) prevention, denial and conspiracy theories, impact control (social distancing, face masks), relief (increased hospital capacity, improved treatments), to reconstruction with hopefully more successful prevention for any similar future event. The global spread of the COVID-19 pandemic has been compared to a tsunami [5], but instead of a response time of 30 min to 3 h, and a spatial reach of hundreds to thousands of kilometers, both spatial and temporal scales differed. What is similar, however, is that even though the underlying science is pretty clear, the psychological, social, and societal response implies time-lags that prevent adequate responses.

Where 30 years of climate change debate still has not induced an adequate and effective societal response [6], there are many similarities between public issue cycles in natural resources management and those in the current COVID-19 pandemic [7]. Especially for the current 'second wave', the social dimensions of protest and breakdown of collective action are being incorporated in epidemiological models [8]. The ambition of building back better is widely shared, but there is no consensus of what it takes, with

some voices pleading for a stricter segregation of wildlife and humans with hygienic control of agricultural production [9], others for a more resilient, diversity-based form of land use in which human vulnerability is buffered [10,11]. We aim to contribute to this debate by discussing the COVID-19 pandemic as a stress test of social–ecological system resilience, analyzed in several current system analysis frameworks, with consequences for the forest–agriculture interface.

An assessment in March 2020 by Petropoulos and Makridakis [12] acknowledged that the risks were far from symmetric as underestimating its spread like a pandemic and not doing enough to contain it would be much more severe than overspending and being overly careful if it would not be needed. The time it took for national governments to move on from denial and conspiracy theories became closely linked to the effectiveness of control and the nature of governance systems [13,14]. With such a pandemic, the pressure on social–ecological systems is mounting considering the challenges being faced through climate change risks which where little progress has been achieved in reducing vulnerability to those risks either [15]. There thus appears to be a clear link to styles of decision making and governance. The aim of this manuscript is to elaborate on the stress the COVID-19 pandemic poses on social–ecological systems and provide insights on how the building back better process could be framed to respond to the multiple challenges already facing social–ecological systems e.g., climate change, ecosystem degradation, poor natural resources governance, among others. The study uses a review of existing published materials and cases and collates the information so that the stress posed by the pandemic is properly understood and taken into account for proper design of effective building back better process. The scope of the study is global but with particular emphasis on tropical and subtropical developing countries.

## 2. Methods

### 2.1. Defining the Analytical Scope

Agroecosystems as part of social–ecological systems, serve as the main livelihood basis for millions while also being the fall back resources at the time of shocks for the majority. The ecosystem services generated from agroecosystems form the backbone of the livelihood of the residents who depend on it. Hence, the resilience of a society and the ecosystem is dependent on how well the ecosystem is managed, which unfortunately is not in a good state in many developing nations widely affected by the COVID-19 pandemic.

One major determinant of the ecosystem services generation potential of a given agroecosystem is land use choices. Land use, as part of natural resource management, can be described as the interface of an 'allocation' choice over four broad categories (Figure 1) and a spatial patterning of 'grain' size of the mosaic. For instance, agroforestry as a concept is associated with both a 'partial tree cover' category between open-field agriculture and closed canopy forest [16], and with a relatively fine-grained ('integrated', multifunctional) mosaic landscape [17–19]. Ecosystem services can, in this context, be understood as direct benefits to humans from (1) natural systems, (2) half-open land use systems, (3) open-field agriculture, and indirect benefits (4–6) derived from landscape scale interactions between land cover types [20], see Figure 1. Management of such landscapes and social–ecological systems should conserve existing natural ecosystems and restore the degraded ones for the benefit of both humans and biological diversity. 'Restoration' in this context needs to consider both the pattern and overall tree cover [21]. In the context of zoonoses such as the COVID-19 pandemic, both the level of 'integration' vs. 'segregation' of the landscape mosaic (influencing initial human or livestock infection risk), as well as the overall cover fractions are likely to be important.

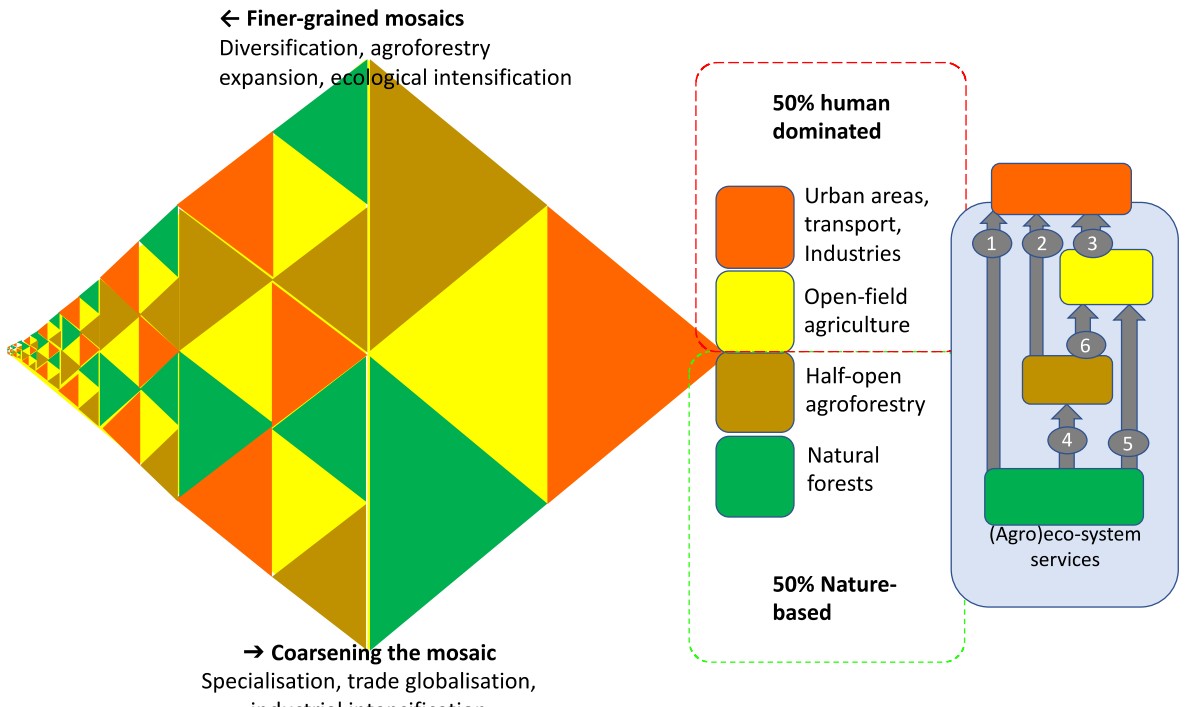

**Figure 1.** Relative composition and spatial pattern (grainsize of the mosaic) as two aspects of land cover at the forestry-agricultural–urban interfaces that jointly determine a set of six (agro) ecosystem services via direct plus indirect benefits people obtain.

### 2.2. Selection of the Relevant Frameworks

As a key step in the methodology, we considered a number of conceptual framings of social–ecological systems subject to disruptive changes. As there are both gaps and overlaps between alternative ways of framing, we settled on three frameworks that allow a multiscaled analysis of proximate and underlying causes, social responses that are informed by but not synonymous with stakeholder understanding of the ecological/medical aspects and offer sufficiently generic perspectives on system-level responses.

For analyzing the depth of the likely impacts and consequences of the current pandemic and the options for 'never waste a crisis' [22] in building better futures, we considered two frameworks used in recent literature. Each has its specific strengths and weaknesses, and a combination, rather than single framing appeared to be attractive. As the COVID-19 crisis affects the balance between short, medium, and longer-term goals in the 'safe space for humanity', the temporary dynamics is an issue that started from initial denial and conspiracy theories moving towards acceptance and searches for solutions. It also showcases the need for a balance between responses along a driver-pressure-system state-impacts responses continuum. In line with these arguments, we specifically explored:

- The adaptive (r-K-$\Omega$-$\alpha$) cycle introduced by the Resilience Alliance [23] (Figure 2A), accounting for the 'collapse' phase ($\Omega$) where existing assets as well as network connectivity are lost, preceding a reorganization phase ($\alpha$), a rapid growth or pioneer phase (r), and a gradual approach to carrying capacity (K). This framework highlights the processes or phases in which change is happening a given agroecosystem once disruption happen.
- Interacting Driver-Pressure-System(state)-Impacts-Responses (DPSIR) cycles [24], with adaptive, mitigative, transformative, and reimaginative responses, advanced by issue attention cycle that encompasses agenda setting, better understanding, commitment and coalitions for change, devolved details of implementation and evaluation of effectiveness (Figure 2B). DPSIR identifies how a given driver, that causes disruption

to the way agroecosystems function, creates pressure and hence influences the system as a whole.

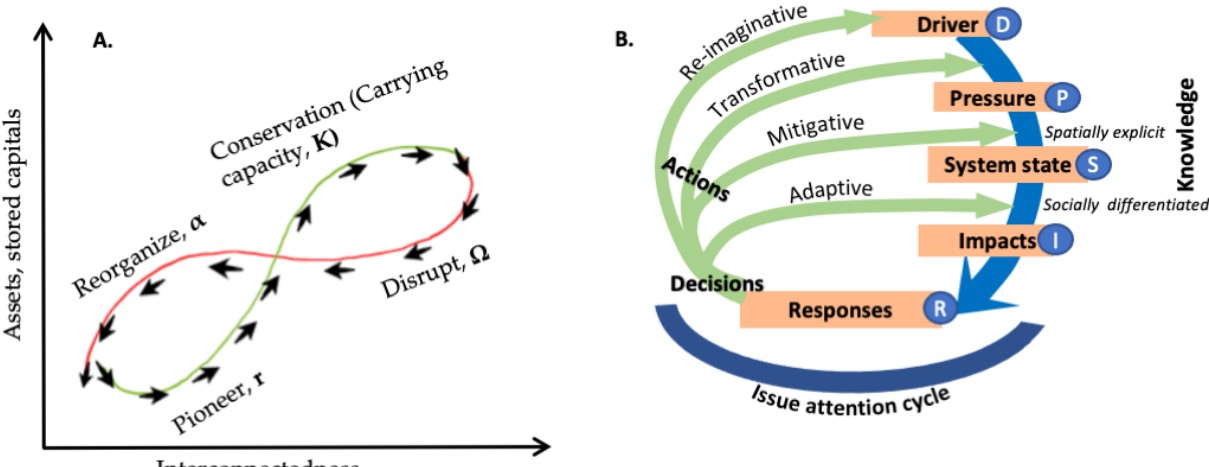

**Figure 2.** Two conceptual schemes that can add insights to possible responses to the COVID-19 pandemic in agroecosystems: (**A**) resilience panarchy [23]; (**B**) connection between Driver-Pressure-System-Impacts-Responses scheme and societal decision making [24].

Each of the framings is built on structure (elements in the system) and function (interactions, cause–effect relations) that we sought to apply to the COVID-19 pandemic by using recent peer-reviewed and other literature related to COVID-19 issues. Often this led to further 'snowballing' to describe and to understand the underlying issues in their spatial, social, and ecological contexts. The purpose of using the two frameworks is to provide a structured logic that is grounded on relevant theoretical frameworks such that the ongoing emergent issues due to COVID-19 could be explained by using relevant conceptual frameworks. The results in this paper will be presented and discussed along the panarchy line (Figure 2A) as a guiding one using the Driver-Pressure-System-Impacts-Responses scheme (Figure 2B) as an explainer for COVID-19 impacts.

### 2.3. Literature Search and Data Synthesis Procedures

The first step in the literature search was to define the key terms that are relevant for the analysis. Table 1 presents the search terms combinations used and the number of scientific publications we found in the literature database. We are well aware that more studies may come out in the future and hence what we present here are the emergent system perspectives as 'results', such that it can inform the ongoing discussion around building better responses by policy makers, academicians, and other practitioners.

We restricted the search database to the Web of Science (Web of Science™) as it is one of the most recognized repositories of scholarly scientific works from around the globe with over 171 million records. The search focused first on the title of the documents. For each document screened, we evaluated its thematic relevance to COVID-19 impacts in relation to agroecosystems. Documents that are not published in open access mode and those that are not accessible through institutional subscription were eliminated as we needed to read the full document before making conclusions on the content of the document.

**Table 1.** Search combinations used for screening literature.

| Search Description | Search Term Combinations (Indexes: SCI-EXPANDED, SSCI, A&HCI, ESCI.) for All Years | Retrieved Documents as of 6 December 2020 | Documents Screened for Further Analysis after Qualifying the Retention Criteria |
|---|---|---|---|
| COVID-19 and its relations to environment | title: (COVID *) AND title: (environment *) | 279 | 123 |
| COVID-19 and its relations to wildlife | title: (COVID *) AND title: (wildlife *) | 18 | 17 |
| COVID-19 and its relations to income | title: (COVID *) AND title: (income *) | 118 | 114 |
| COVID-19 and its relations to migration | title: (COVID *) AND title: ('migration') | 36 | 29 |
| COVID-19 and its relations to livelihood | title: (COVID *) AND title: (livelihood *) | 9 | 7 |
| COVID-19 and its relations to employment | title: (COVID *) AND title: (employment *) | 29 | 24 |
| COVID-19 and its relations to social capital | title: (COVID *) AND title: (social capital *) | 13 | 13 |
| | Total | 502 | 327 |

Note: The search terms combinations indicated uses the Boolean search approach with AND, OR, NOT. * denotes that the search captures any word that has the root word that precedes the *.

Based on the above two criteria i.e., relevance to COVID-19 and agroecosystem attributes and access to the document, of the 502 documents screened from the Web of Science portal, only 65% fulfilled the retention criteria. The highest rate of elimination occurred on the search combination for relations between COVID-19 and Environment with less than 44% retained as most of the document bearing that word were linking it to the laboratory environment and other environmental factors (e.g., temperature and humidity) widely used in relation to the COVID-19 context and hence eliminated from the current analysis.

For each document, the stated impact on the selected attributes (see Table 1) were assessed and the impacts were categorized as positive, negative, or both; as there was not standardization in quantifying impacts, we had to stay at a qualitative level for the current screening. Documents presenting both positive and negative aspects were categorized under both, those only talking about the negative impacts on selected attributes were clustered as negative. A similar approach is used for those stating positive impacts. See supplementary material 1 for the spreadsheet presenting the analytical dataset from literature.

## 3. Results and Discussion

### 3.1. Summary of the Impact Reportings of Effects of the COVID-19 Pandemic on Social–Ecological Systems Attributes

Of the 327 documents reviewed, 43.73% of them clearly stated COVID-19 had negative impacts on social–ecological systems, while 30.89% presented both positive and negative impacts. The main negative aspects reported are along the lines of social aspects of social–ecological elements such as income, livelihood, employment, migration, etc. It is also important to note that positive effects of COVID-19 were reported by about a quarter of the reviewed papers, largely along the reduced human impacts on the environment due to travel and movement restrictions. Detailed breakdown is presented in Figure 3.

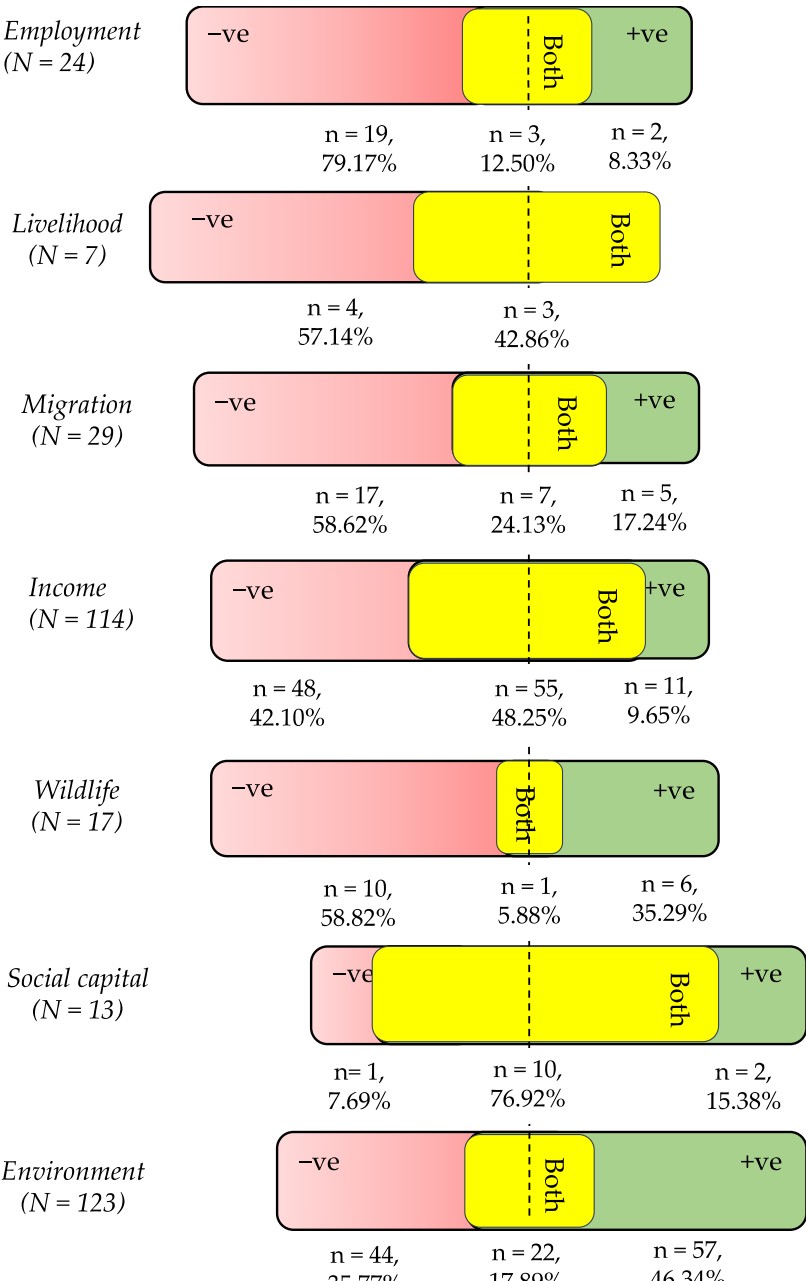

**Figure 3.** Impact potentials of COVID-19 pandemic on selected social–ecological systems attributes. Note: +ve—positive impacts of COVID-19; −ve—negative impacts of COVID-19. *N* stands for the number of valid documents used in the review process for each category. The broken line divides the yellow box representing both in half.

Results from the assessments of the positives and negatives ratios (which were computed by assigning +1 for any document reporting positive impacts and −1 for any document reporting negative impacts) gave a clearer picture of the impact of COVID-19. It is evident that, compared to the other attributes analyzed, the large positive shares reported are mostly for *environment* and *wildlife* management.

COVID-19 has shocked the world infecting millions of people [25] and causing total disruptions in how the planet is functioning and operating [26,27]. Due to the disease, countries have closed land borders, ports, and even their airspaces except for emergencies or supplies of medical goods and equipment. At a time when the planet is moving to a more globalized scheme, these measures, adopted to safeguard their population and

also to contain the virus spread, created shocks to the broader economy, livelihoods, and societal networks [28]. This resulted in significant social effects that created further stress to the already prevailing challenges of climate change, environmental degradation, and other livelihood affecting causes. Results in Figure 3 showed the impacts of the disease on income, employment, migration, and livelihood in general.

Though COVID-19 effects were global, developing countries were the most affected due to disruptions of economic activities including production and export trade [29]. Most of these countries rely on agroecosystems that are sources of agricultural products for export. The sustainability of the agricultural sector and its agriculture-led industrialization (e.g., in Ethiopia—coffee and flower; Kenya—tea, avocado; Ghana, Cameroon, and Cote d'Ivoire—cocoa, etc.) entirely depends on how the components of the agroecosystems (i.e., land, water, biodiversity, etc.) are properly and sustainably managed. The pandemic exposed the loopholes in the highly advocated export market by revealing the weak readiness countries have when such global issues affecting larger population arise.

Pre-COVID-19, products were exported 'immediately' to countries of destination for further processing, consumption, or utilization. Hence, producers and exporters did not need an extensive level of preparedness in terms of storage places (e.g., cold rooms, packaging, etc.) and raw materials supplies from the producer communities were more or less secure as far as production inputs are supplied, and no major climate influence occurred. However, during and after COVID-19 (only for some countries) those operating on perishable agricultural products for international markets (such as tomato, flowers, vegetables, avocado, and other crop products) are counting losses as the uptake in the importing countries is disrupted due to the lockdown. Overall, the net gains from agricultural products trade faced an uphill challenge that kept the sector in limbo. The economic impacts trickle down to the individual households who are the leading producers of the agricultural products marketed and exported. The pandemic exposed the vulnerability of the globalized trade infrastructure that was setup for decades in a bid to link the small-scale producers with the exporters and processors and the global market often located in the developed nations.

The other side of the coin has been a marked improvement of air quality especially the outdoor one [30], reduced greenhouse gas emissions, and an 'anthropause' that provided opportunities for wildlife to reclaim part of the space appropriated by humans [31]. Aspects of river quality were also recorded to be improved [32], but the single-use plastics created additional waste problems [33]. The uncontrolled disposal of the facemasks and other COVID-19 kits were, however, expected to increase the level of water pollution.

*3.2. Collapse Phase and Early Signs of Reorganization*

3.2.1. The Pandemic Effects on Communities and Early Responses

For the immediate impacts of the pandemic and lockdown measures, the collapse ($\Omega$) phase of the 'panarchy' loops (Figure 2A) suggests that loss of assets accompanied the loss of connectivity in cascading cause–effect chains. The focus in the early phase of the pandemic has been about controlling the spread of the virus and saving lives of those affected, but cascading effects may have made the negative impacts larger than foreseen. The trickling down effect of the virus soon had devastating impacts on broader natural resources management issues. That, in turn, will most likely shake the whole economic base of most developing economies that rely on natural resources and its goods and services.

The influences of the pandemic are either direct (e.g., through job losses, income decline, etc.) or indirect through the consequences of the pandemic on economic activities, production systems, and supply chains [34]. Apart from this pandemic, there was no recent record of countries closing their air, land, and seaports except for military or political reasons. The COVID-19 pandemic made countries respond aggressively to save lives over livelihoods and focused on health over wealth. This measure, however, came with a number of negative consequences. The sectors that used to support livelihoods suffered significantly. Economies shrank, jobs were lost, and people were made to venture into

unsustainable and dissatisfying activities e.g., exploiting natural ecosystems that are the basis of their current and future generations. The response strategies adopted varied by the scale at which the responses are looked at. The national and subnational level responses focused on minimizing movements of people in and between countries and also restricting or totally blocking entry of foreign nationals into the country as a whole. In Africa, for instance, as of April 2020, 38% of the international destinations imposed a total or partial closure and 45% of the destinations restricting direct flights [35]. This almost crippled the tourism and hospitality sectors [36] resulting in employees in these sectors being laid off. The effect for some countries is so severe due to the very high dependency on international tourism as the backbone of their economy. For instance, for some countries, tourism accounts for over 50% of their total exports (Cabo Verde—67%; Sao Tome and Principe—65%; Gambia—53%; Comoros—51%). Others such as Seychelles (38%), Mauritius (34%), and Tanzania (29%) also heavily depend on these sectors [34]. UNWTO [37] also indicated that international tourist arrivals for May 2020 shrank by 98% in sub-Saharan Africa compared to May 2019 figures.

At the community level, responses varied by the types of community capacity and livelihood sources mainly. For instance, with the closure of manufactories, many casual workers became jobless and had to return to or had to send their immediate families to their rural parents and relatives. The rural–urban migration took its opposite and urban–rural migration [38] became dominant, even though temporary. In any case upon arrival, such returnees have to cater for their families and the tendency to engage in the collection of wild foods and hunting to feed their families becoming common. Clearing forests for timber and charcoal production (in dryland areas) to generate income becomes an immediate measure at the forest margins, even aggravated by the laxity in low enforcement as broadly discussed in later parts of this paper. Table 2 provides insights into some of the predominant responses commonly mentioned.

**Table 2.** Characterizing the responses to the pandemic at the household level in the African context.

| Types of Responses | Main Drivers of Response in Relation to COVID-19 | Consequences on Agroecosystems |
|---|---|---|
| Full family relocation (urban to rural; rural to rural) | Unemployment; income shortage | Increasing the need for food, feed, water, and other basic needs in the agroecosystems |
| Partial family relocation (urban to rural; rural to rural) | Either the husband or wife is left in urban areas with others relocating to rural areas | Same effect as above |
| Food and other consumable support (rural to urban) | Rural families send food and other supplies to their relatives in urban areas | Same effect as above |
| Unemployed members of family relocation (urban to rural; rural to rural) | Loss of jobs by youth members of the family and relocating to rural homes | Increased rate of forest clearance, poaching, charcoal production |
| Changing livelihood means | Loss of jobs in rural economic sectors e.g., tourism and hospitality, manufacturing, export sector slowdowns, etc. [35] | Same as above |

### 3.2.2. Likely Effects of COVID-19 Pandemic on Households and Local Livelihoods Dependent on Agroecosystems

Livelihood in the developing world is strongly dependent on ecosystem services derived from agroecosystems. The COVID-19 pandemic has impacts on needs fulfilled from agroecosystems both in the short-term and long term. The short-term impacts are largely through its influences on the supply of food and energy, generation of income and health benefits. Table 3 describes the impacts and the implications of COVID-19 on these benefits.

**Table 3.** Benefits from agroecosystems and how COVID-19 influences them.

| Needs Categories | Sources from Agroecosystems/ Socioecological Systems | How COVID-19 May Affect the Supply Sources |
|---|---|---|
| Food | Crops | Due to disruptions in inputs delivery and crop calendar, crop production is likely to be affected [39]. The need for additional food for family members who lost their job in urban and service sectors increases. |
| | Livestock | Due to disruptions in livestock market and medication particularly vaccinations implying decline in productivity of livestock and in the medium-term decline in herd sizes. |
| | Wild foods (fish, bushmeat, honey, tree foods) | Increased extraction for wild foods due to market supply of food crops disruptions. People engage in hunting and wild food collection increasing pressure [40–42]. |
| Energy | Electricity | Less income for households [29] means declining affordability of electricity. |
| | Forest wood | Extraction of wood from forests for energy may increase due to the low affordability of electricity. |
| | On-farm trees | Use of on-farm trees for cooking and heating increases [30]. Farmers may even sell wood for energy to earn income. |
| | Solar power | Access to solar panels and other accessories declines due to international trade disruptions [43]. |
| Water | Freshwaters (stream sand rivers) | There is a rising waste management problem, especially for face masks and gloves produced, used and disposed ending up in streams and rivers polluting water sources [44–46]. |
| Health | Local health facilities | Access to local health services declines due to shrinking income and people revert to traditional and herbal medicines. People are also scared to go to health services and more relying on traditional sources of medicines mostly from the agroecosystems. |
| | Herbal/traditional medicines | Exploitation of plant-based traditional medicines increases as affordability of health facilities declines. |
| Income | Annual crops | Sales of crop harvests disrupted [29,47] and those producing fragile crops lose significantly. Input supply disruptions also mean a likely decline in productivity. This affects the farmers' income [48]. |
| | Tree commodities (coffee, coconut, oil palm, etc.) | Tree products marketing reduced due to export disruptions and market slowdown [38]; input supplies for tree crop farm management disrupted implying poor tree farms management; labor shortage as laborers were returning to their families in remote or rural areas. |
| | Shrinking employment opportunities | Activity slowdowns or closures led to job losses by those working as casuals and in the service sector workers in urban and rural areas [47,49,50]. |

The long-term impacts of COVID-19 could probably be far wider than anticipated though it could be difficult to authoritatively quantify at the moment.

In contrast to many 'natural disasters' where the loss of assets is the primary trigger of a collapse (Ω) phase, the COVID-19 pandemic affected human livelihoods primarily through the 'lockdown' measures taken to control the spread of the virus (Figure 4). Once the cascade took on its own dynamic, however, shifts occurred on both assets (*y*-axis) and interconnectedness (*x*-axis).

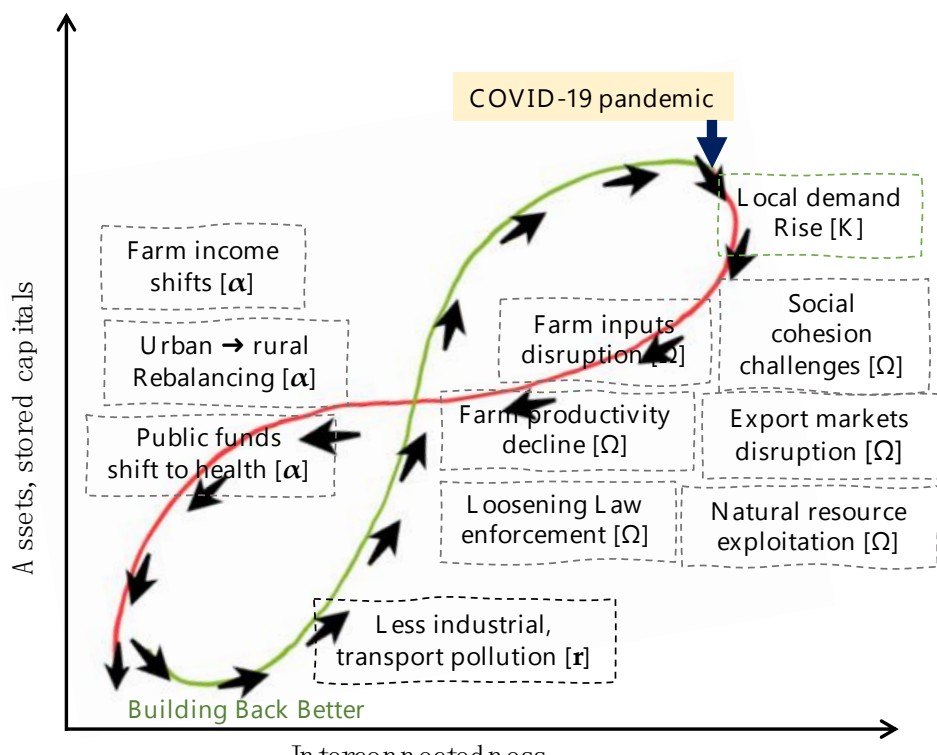

**Figure 4.** Examples of the cascading events induced by the lockdown and disruption of connectivity, triggering loss of assets featuring observed phenomena in relation to COVID-19 with some prominent emerging positive and negative aspects along the resilience panarchy framing.

### 3.2.3. COVID-19 Induced Social Support System Dynamics and the Resulting Pressure on Agroecosystems

While the collapse phase may have seemed to be a free fall, social safety-nets came into play and provided opportunities for early steps of bouncing back. Developing countries, in particular sub-Saharan Africa, experienced a significant rural-to-urban movement in the last few decades leading to swelling of the urban and suburban areas strongly dependent on the rural agroecosystems. The migration was largely driven by the opportunities of employment either as casual or other forms of employment. Hence, almost the majority of households have one or more members of the family who have moved to urban areas for gaining employment of any sort. With the emergence of COVID-19, and measures taken to curb the spread, many employers laid off and/or reduced staff and laborers, leading to job losses. The urban counterparts, since they lost their jobs and had no income sources were asking for help from relatives and family members in rural areas. Others who lost their jobs returned to rural areas, thus increasing demands for consumables in rural households. This may increase the need for farmlands which is often gained through expansion into forest and woodlands especially for those living at forest margins.

Hunting for bushmeat and collection of wild foods (e.g., wild fruits, wild vegetables, honey, eggs, etc.) were seen as easy ways of complementing the rising food needs due to increasing number of people in a family due to the urban returnees. Reports of increasing exploitation of forest and woodland resources was reported in many cases [47]. Those who had families in rural areas and used to work in urban areas have to now generate income to sustain their families. With loose forest and wildlife protection rule enforcements due to the pandemic, illegal exploitation of forests and wildlife increased mainly to produce locally marketable products such as charcoal, timber, and/or bush meat.

The pressure is not only from those laid off from urban areas but also from the member of local communities serving in the service sector in rural contexts such as tourism,

factories, and infrastructure constructions. There was a mass return to the rural areas where the source of any basic need was largely confined to what the agroecosystems could generate. Overall, agroecosystems became the fallback options to cope with the impacts of the COVID-19 pandemic, at least in the short-term. As a result, the pressure on the agroecosystem rose thanks to the extractive pressure for various household needs.

### 3.3. COVID-19 and Its Potential Impacts on Sectors Directly Linked to Agroecosystems

3.3.1. Broader Sector-Level Issues Arising Due to COVID-19 and the Consequences

COVID-19 also affects a number of sectors that directly rely on agroecosystems. Table 4 summarizes the anticipated or observed impacts of the virus. These impacts, in turn, affect rural households in different ways leading to far-reaching impacts.

**Table 4.** Impact pathways of COVID-19 on sectors strongly linked to agroecosystems and broader natural resources.

| Sector | Immediate Issues/Impacts | Likely Consequences |
|---|---|---|
| Agriculture | Decreased labor supply (even if it may be temporary) | Poor agricultural productivity [51–53] |
| | Limits farming input supplies | Total production loss or low agricultural production and hence poverty [53–56] |
| | Increased costs of agricultural inputs | Reduced benefits for farming communities [45,56] |
| | Weakens innovation through research for development | Future of agriculture is uncertain unless innovation picks up |
| | Less adaptive crop varieties are used for farming | No or very low productivity with likely effect of food insecurity [29,56] |
| | Less disease and pest resistant crops varieties used | Poor agricultural productivity and likely food insecurity [45] |
| | Poor soil management | Degradation of agricultural lands and hence low productivity [52,57] |
| Forestry | Illegal forestland encroachment expansion [30] | Degraded forests and hence low ecosystem services generation e.g., hydrological and habitat values |
| | Any forest inventories may not happen | Except delays in forest data though not an immediate worry |
| Fishery | Unpredictable fishing livelihoods | Future fishing dependent livelihoods may be threatened [39] |
| Wildlife | Poaching and bushmeat hunting expansion [48] | Wildlife resources may decline |
| | Exposure of protected areas and conservation areas to intruders [39,47] | Conservation areas could be exposed to land grabbing and wildlife could be endangered |
| Water resources | Pollution of water resources may be a challenge [42–44] | Community health impact may become a significant concern |

Overall, protected areas in developing countries are under pressure due to the need for utilization by local communities. So far, the most effective mechanism to safeguard such resources is deploying a large number of rangers and guards who man the areas either by using technology or manual means.

With COVID-19, there are reports of a rising amount of illegal access and use, particularly poaching and deforestation. Many attributed the rise to the shrinking operating budgets of institutions that used to take care of such resources. The National Geographic, in its latest report [58], revealed that poaching is becoming a major threat to the protected areas due to the resource limitation due to COVID-19 resulting in low returns, meaning less rangers manning the parks and conservancies. ABC News [59] also echoed the same issue referring to the situation in Kenya where experts voiced concerns over the rising poaching [60] and deforestation. As Bates et al. [61] hypothesized the decreased global

mobility may reduce the pressure on biodiversity and may even increase sightings of wild animals in urban areas [62].

The loosening of restrictions are a major concern for the African continent where there is a very high rate of bush meat hunting and consumption, especially at this time when there are limited income sources to buy food from the market due to people not having access to markets as well as lack of jobs to earn money. The Telegraph [63] released a bulletin, based on the views of experts, warning a significant surge in bush meat hunting in the continent which poses a major threat to wild animals. This may even increase the chances of another zoonotic disease outbreak depending on the nature of the animals hunted and consumed. Unless swift measures are taken, the declining number of rangers and personnel used to protect such areas exposes the wild animals and the woodlands to severe degradation. This may have a trickling down effect diminishing the sustainability of the landscapes and then the livability of the area.

### 3.3.2. Disruptions of Interventions and Innovations for Agroecosystems Management

Three main pathways of impact can be identified here—activity slowdown, activity discontinuation, and limited opportunities for innovation. With countries declaring lockdowns in many parts of the globe, movement of expert personnel who could have moved from place to place to implement activities was severely affected. As a result, most of the agroecosystem management activities that could not be performed via remote connections have been put on hold for many months. This movement restriction has mostly affected the research for the development aspect of natural resources' management interventions. This slows down innovations that could have tackled problems being faced in the most affected sectors such as agriculture, forestry, fishery, water resources management, wildlife, and mining. Most field operations were suspended or were going on with minimal engagement on the ground. This has delayed the planned implementation of many interventions that could have contributed significantly to the productivity improvement, disease control, and/or site management interventions in these sectors. Zellmer et al. [62] reported cases of researchers and scientists indicating research activities getting discontinued due to the pandemic and hence affecting finding solutions to biodiversity challenges. Manenti et al. [64] also reported that the lockdown had beneficial effects on biodiversity conservation, especially citing a decline in road killings of wild animals in Europe. Paital [65] also reported increase sightings of animals due to the lockdown. For urban contexts, such positive aspects in relation to wildlife benefits are very much welcome news. However, in contexts where wildlife is seen as source of food for the community, and where low enforcement weakens due to inadequate manpower to take care of the job, the lockdown could even expose wildlife to greater danger.

In agriculture, forestry, and fishery sectors, most interventions are time sensitive i.e., season-dependent and if the schedule is missed, one has to wait for the next year to implement similar tasks. For instance, coffee farmers have to start preparing coffee fields in the month of January to March, months when most coffee producing countries went into a lockdown. The same is true with disease control measures for coffee. For coffee cherries that will be collected the next coffee season, the above activities that should have happened during the lockdown are crucial. If those activities do not take place, coffee production is likely to be affected significantly the next season. The impact is largely due to dependency of the activities on labor availability, which is not a problem when movement restrictions were not in place. However, during the lockdown, no movement was taking place as people were concerned about their own families and community's health.

Due to movement restrictions, field inventories, surveys, data collection, and other relevant field activities [66] were slowed down or discontinued completely to avoid risks to the personnel and community within which activities are to take place. Manenti et al. [64], using responses from protected area managers, found that there was a challenge to implement activities to manage ecosystems due to the lockdown leading to flourishing of

invasive species which are managed through human interventions during normal times when access is not limited.

At farmers level, the impacts are far reaching. For instance, due to the nonessential travel and movement restrictions and the lack of prior preparation, farmers could not access input supplies such as fertilizers, disease and pest control inputs, improved seeds, etc., which are crucial for farmers to secure or maintain agricultural productivity to earn income and to fulfill the needs of immediate family.

### 3.3.3. Shrinking Returns and Revenues from Agroecosystem Dependent Sectors

With the disruptions in global trade and local movements, trade of forest and agricultural products has been affected. With agriculture being the backbone of the economy i.e., up to 50% of the GDP in sub-Saharan African countries [67] and most countries taking the export-led agricultural development schemes, the disruption led to significant losses in returns. The damage is so severe for those engaged in perishable products such as vegetables, avocado, and other products such as milk and milk products. The loss is also magnified by the lack of preparedness for such an unforeseen event that no one expected could happen to disrupt the import–export schemes.

The closure of airspaces, seaports, and land borders led those engaged in exporting agricultural commodities to be reluctant to collect the products from the farmers who primarily produce the agricultural commodities. This implies that farmers are the ones to incur significant losses in such contexts because they invest their time, labor, and money to produce those products. In some countries, even movement of goods was also restricted thus affecting the producers even more.

One vital sector that usually generates substantive revenue for natural resources management (NRM) sector in a number of countries is tourism. In many African countries, the sector is strongly dependent on the ecosystem i.e., flora, fauna, and the landscape. For instance, in East Africa the major tourist attractions are wildlife and natural ecosystems. With the movement restrictions, people stopped the cross-country movement of people, resulting in tourists abandoning the region [48], though temporarily. With the lockdown, revenue from the sector has shrunk significantly. It is important to note that the tourism sector supports most of the wildlife parks, sanctuaries, and even the private parks in Africa. For example, UNWTO [35] indicated that as of April 2020, almost half of the global tourist destinations have closed their borders either totally or partially. In Africa, where international tourists make up the biggest share of the tourist industry, the impact on the revenue is very high. For instance, almost 20% of the African countries depend on tourism with the sector accounting for at least 20% of their exports. The local economy is also largely affected because of declines in international tourist arrivals. International tourists account for at least 40% of tourists in 13 out of 54 countries in Africa [38] and the impact this will have on the sector and those working in the sector is very strong.

With the shrank revenue due to COVID-19, most of the natural resources (wildlife, landscapes, and other natural habitats) that the sector relied on received limited management investments [47] due to resource scarcity. Unless there is a new support scheme these resources may face significant degradation due to lack of effective management till the sector recovers after COVID-19. It is unfortunate that the countries where such resources are located are also facing financial constraints forcing them to channel available resources to priority and urgent interventions to control COVID-19. For example, Figure 5 shows a schematic of the interconnected impacts of the pandemic on wildlife conservation.

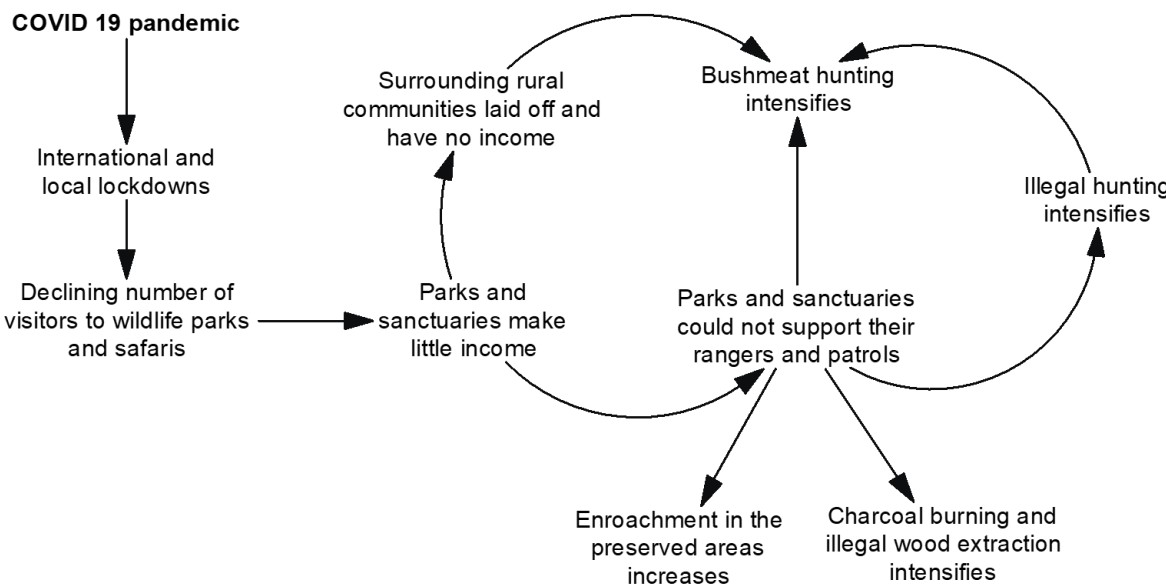

**Figure 5.** Schematic of how COVID-19 affects wildlife conservation in a broader context [38,47,48].

When such sectors suffer economically, the local communities who depend on them also suffer significantly. They may then begin to look for alternative livelihood means such as wood harvesting, charcoal production and sale, illegal fishing and hunting, and others, which in turn exacerbates natural resource degradation and thus threatening the sustainable management of the area at large.

### 3.3.4. Shrinking Social Capital for Agroecosystem Management

Natural resources management in Africa is anchored to the social support systems that are built within the communities. The labor divisions observed are typical of how the social system is configured to ensure that every household member has a contribution while also neighbors, whether relatives or not, lend hands to one another to ensure everyone achieves the goal of production, conservation, construction, etc. The role of such social systems is really becoming very important with most of the youth preferring to migrate to urban areas and other countries even. For instance, in Ethiopia, communal labor support groups ('Daboo' and 'Daadoo') [68] are typically set up to assist neighbors in agriculture, forest management, coffee farming, house construction, etc. The support group is common among Oromo communities in particular, and such social structures were significantly affected by the COVID-19 emergence as members got concerned about their safety.

In this regard, the impact of COVID-19 is largely due to the restriction in social interactions which communities were advised not to practice until the situation of the pandemic is kept under control. Communities therefore are not even getting together for social support despite the dire need for such local mechanisms that facilitate the supply of the scarce labor for agriculture and other natural resource management interventions. Community-level planning processes to address natural resource management challenges are also put on hold. This may lead to significant degradation of natural resources, particularly areas that ought to have been managed in the surrounding landscapes.

### 3.3.5. Diversion of Resources away from Agroecosystems to Pandemic Response

During the pandemic period, countries with limited financial resources are taking drastic measures by reallocating resources to tackle COVID-19 [69] and limiting resources that go to other sectors. Sectors included in NRM such as agriculture, forestry, fishery, tourism, energy, etc., are often left with minimal investment from the government resources and left to any support that could be obtained through bilateral/multilateral/private sector supports (Figure 6). However, Maher et al. [70] disclosed that even nonprofit

organizations are experiencing shortfalls in their financial status. In another study from the USA, Clemens and Veuger [71] revealed that COVID-19 impacts government revenues as economic activities are slowed down. A similar fiscal challenge was reported for Italy too [72].

Despite being the critical pillar of the economies of many developing nations, natural resources management efforts are often given low priority in resource allocation and deployment of competent human resources and infrastructure (see Zimmerman et al. [73] for the case of agriculture for example). Where there are emergency situations, past experiences reveal that budgetary allocation for natural resources management such as forests, wildlife, etc., are the first casualties to be trimmed down. Surprising even, in many developing countries (especially in Africa), resource allocation for agricultural resources management is very small compared to other sectors despite NRM making up the major share of their gross domestic production. In sub-Saharan Africa, the sector makes almost 15% of the GDP (with the smallest being 3% in Botswana to about 50% in Chad) [67]. Agriculture cannot survive if the soil it is produced on is not conserved well both on-farm and off-farm. It cannot continue sustainably if water resources in the surrounding areas dry up and water supply for animals is not secured. All this can only be possible if resources are allocated for such interventions.

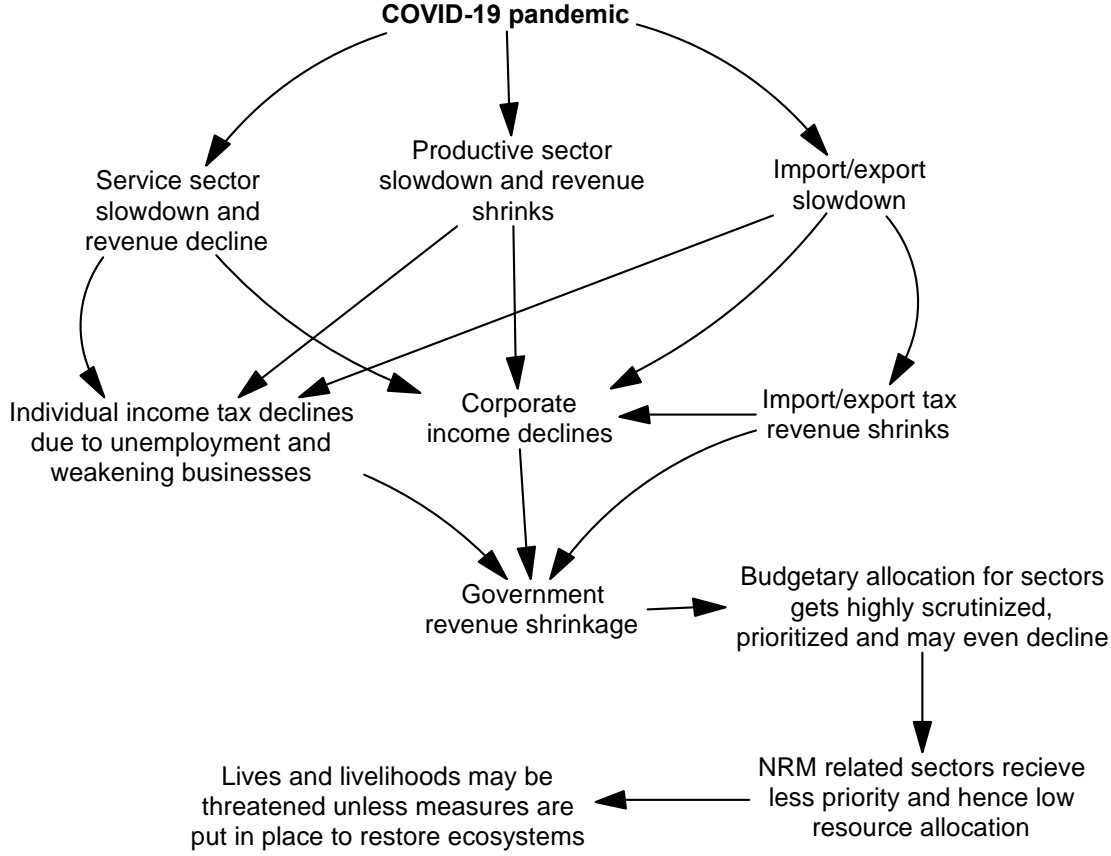

**Figure 6.** Impact pathway linking government revenues and natural resource management (NRM) activities [74].

The shifting priorities challenge is not only at the national level or government level. It is also being reflected temporarily among international donors and lenders that used to support natural resources management [75]. It is legitimate that more resources should be allocated to tackling COVID-19-related challenges, but if the shift is not done in an informed manner, the life support of humanity could fall into serious danger and it may not be that quick or easy to fix the problem. This requires serious consideration of sustainability

context such that the balance between today and tomorrow can be made a rational decision-making process.

## 4. What Ways Forward?

### 4.1. Building Better Futures: Broader Response Options at Socioecological Scale

Though the answer to the question 'how do we build back better and stronger' may not be that easy to answer as the knowledge about the impacts of the pandemic is getting revealed with time, we have tried to put together our insights on how some potential interventions could help the process better. OECD [76] states that building back better should involve a significant investment in ensuring future shocks are managed properly. At the local level, this needs efforts to avert environmental degradation and deliberate investments in sustainable practices in a context-specific manner to ensure the process is inclusive and incremental to the existing efforts.

The COVID-19 pandemic exposed the vulnerability of the broader agroecosystems related sectors and the local livelihoods to challenges that threaten the livelihood of the society that depends on it both for consumption and income. Addressing these vulnerabilities needs measures that cascade down from the national level to landscape (jurisdictional) and household levels. It needs a concerted effort across scales with decentralized roles and responsibilities for actors critical at the various levels. Table 5 below highlights some proposed measures to ensure the building back better process is more effective and inclusive at the landscape and household levels.

**Table 5.** Potential measures to reduce impacts of COVID-19-like disasters on socioecological systems.

| Areas of Action | Potential Measures for Building Better Futures |
| --- | --- |
| Landscape (jurisdictional) level | Establishing natural resource-based safety-nets through better landscape management, which includes averting ecosystem degradation, ecosystem restoration, waste and pollution management |
| | Enriching landscape-level practice portfolios with climate resilient production systems such as agroforestry |
| | Strengthening enforcement of rules and regulation for sustainable use and management of agroecosystems |
| | Boosting natural resources-based enterprises with strong local demand |
| | Designing cross-jurisdictional resource management strategies to reduce illegal exploitation during fragile moments |
| | Investing in postharvest infrastructures (e.g., cold-rooms, storage units, transportation, etc.) to reduce postharvest loses |
| Household level | Embracing diversified and climate smart production systems that depend less on external inputs. For example, perennial crops are less likely to be affected as compared to annual crops which are season specific |
| | Proper land use plans that promote multifunctional production systems could help in ensuring the sustainable supply of basic needs i.e., food, feed, water, and energy |
| | Embracing climate smart microfarming options in land scarce environments such as urban and periurban areas e.g., fruit trees and perennial vegetables in homesteads |

### 4.2. Mending People–Nature Relations: Management of Socioecological Systems to Build Better Futures

The COVID-19 pandemic can potentially lead to two opposite conclusions with respect to desirable land use: (1) risks of zoonoses that infect humans and/or their livestock may be minimized by hygienic segregation of anything 'wild' and the human spheres of life, but (2) human vulnerability to the lockdown was lower in 'integrated' landscapes where there are multiple livelihood options and low 'fragility' [77].

We can now relate these contradictory effects to other ongoing pressures at the Forest–Agriculture interface (Figure 7A). Among the 17 Sustainable Development Goals, both changes towards more urban and open-field agriculture and changes towards agroforestry and natural forest are desirable (Figure 7B), with a trade-off determined by the terms of the 'Anthropocene equation' [78]. From a strong tradition of high agrobiodiversity at both farm and landscape level, farmers in many parts of the tropics have increasingly 'outsourced' staple food production [79] and relied on market-based income, with an attraction to specialize on the products and market channels that work best for them, including 'traditional foods' appreciated in urban areas that were the likely start of COVID-19. However, shocks are part of the system farmers need to be prepared for and deal with. Part of the shocks farmers face originate in 'natural' disasters such as volcanic eruptions, earthquakes, and tsunamis, although their aftermath is often aggravated by changes in the tourism-related industries [80]. Others originate in human behavior, such as the financial crisis of 2007–2009 or terrorism-related changes in tourism preferences. The COVID-19 crisis started with a pandemic but was modified by an unprecedented 'lockdown' with restrictions on travel as well as long-distance transport within the country and a complete collapse of tourism.

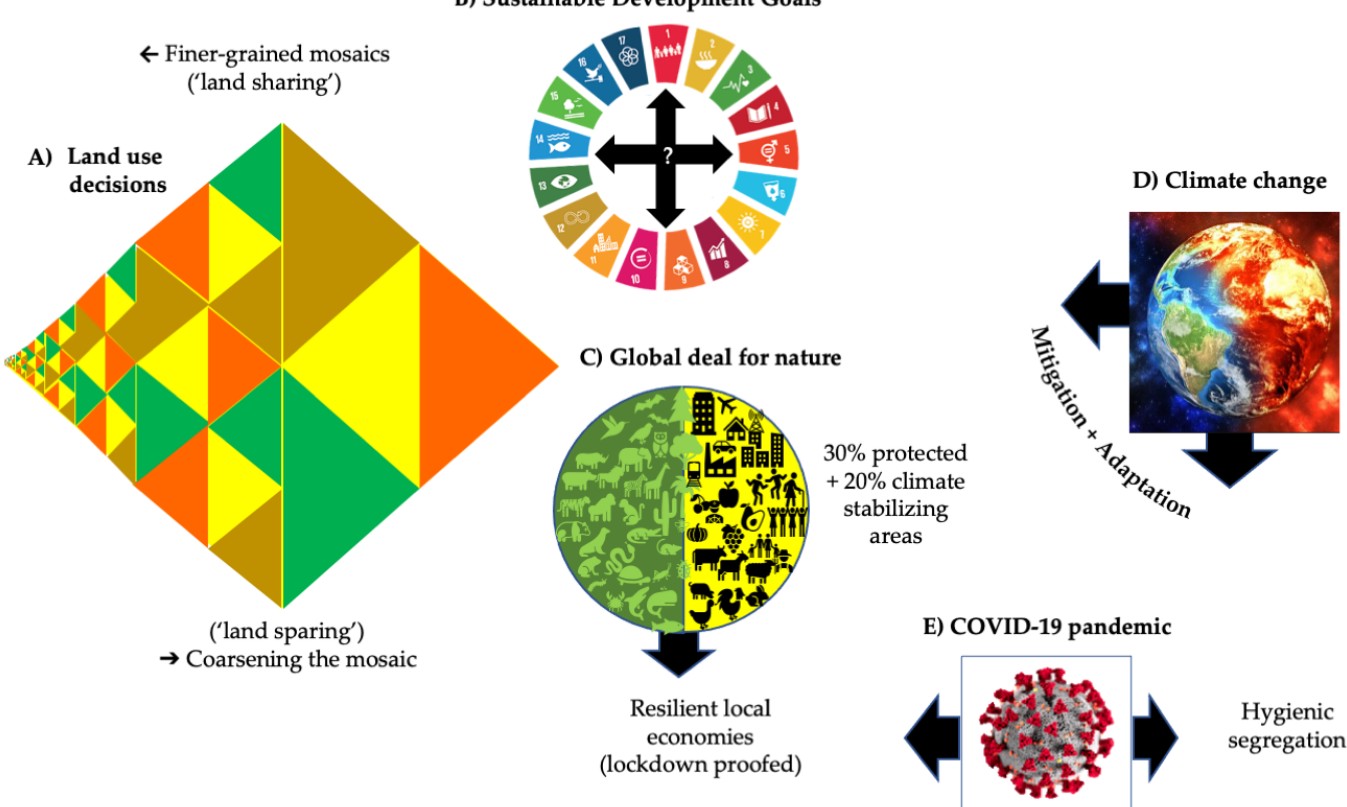

**Figure 7.** Comparing the forces at play in agroecosystems: (**A**) changes in spatial configuration of four land cover types (compare Figure 2); (**B**) the sustainable development goals implying trade-offs in desirable changes in land cover and spatial pattern; (**C**) consequences of the proposed 'global deal for nature'; (**D**) consequences of climate change policies; (**E**) consequences of zoonotic pandemic prevention.

Under the slogan 'nature needs half' [81] a vision ("A world in which people respect nature's needs, and life on Earth flourishes"), has been formulated to turn half of the earth into a series of interconnected protected areas [82,83], expecting that this can conserve about 85% of remaining biodiversity (Figure 7C). According to global data, currently around 15.4% of the earth's terrestrial areas and 3.4% of its oceans has protected area status [84,85], but the 'paper park' without effective protection has not been separated

from these numbers [86]. A proposed 'global deal for nature' modified the proposal to 30% protected areas +20% climate stabilizing areas [87]. This argument has inherited the 'land sparing' perspective that intensifying agriculture increases chances for nature conservation. However, it may be noticeable that the economic arguments that appear to support the 'half earth' perspective [88] are largely based on the expected increase in 'tourism' to nature's half, that may increase rather than decrease zoonosis risks. Counterarguments have challenged that this half earth plan would not meet its conservation objectives but would have widespread negative consequences for human populations in the areas of highest biodiversity value. An alternative radical action may lead to solutions that are both more effective and more equitable, focused directly on the main drivers of biodiversity loss by shifting the global economy from its current foundation in growth while simultaneously redressing inequality [89]. Solutions cannot ignore what happens in 'the other half'. In the ongoing discussions for the post-2020 Agenda for the Convention of Biological Diversity, this has been modified to 30% conventional protected areas and 20% under indigenous and customary rule protection [90].

Meanwhile, the global climate change agenda depends on protecting and restoring terrestrial carbon storage and emission reduction in the agricultural sectors that may imply a shift to more forest and trees, as well as finer-grained mosaics (Figure 7D).

Finally, the counteracting conclusions in zoonotic pandemic prevention of better segregation and increase resilience (Figure 7E) need to be reconciled with the other trends to become part of a reimagined and transformative land use policy agenda.

## 5. Concluding Thoughts

The way the COVID-19 pandemic disrupted human lives and livelihoods proved to be a stress test for the social–ecological systems at the forest–agriculture interface in developing countries, as part of rural–urban systems and the global economy. It has further impacted socioecological systems in countries where weak governance, poverty, and degradation had already weakened the systems. This paper set out to understand the impact pathways in order to improve prospects for building back better in these developing countries. Drawing from multiple perspectives in op-eds, opinion pieces, and articles, we teased out a number of pathways around impacts at multiple scales. We explored sectoral level, landscape level, and household level impacts on ecosystem services as a result of COVID-19. Prospects for building back better were analyzed including across sectors.

It emerges overall that in order to build back better, a holistic conceptual and integrative approach has to be developed both short term and long term to enhance effective, efficient, and equitable sustainable development that would enable a less stressful impact of a COVID-19-like event in the future. Short-term efforts targeting mitigation of impacts could deploy agroecosystem practices at the forest–agriculture margins such as agroforestry and others. The objective being to address direct impacts on food, fiber, and fuel supply chains, as well as the human capital impacts such as jobs, nutrition, social, and income.

In the long-term, it would be important to design and focus on building back better actions around adaptive, transformational, and reimaginative approaches that target system changes. Adaptive approaches need to focus on adjusting socioecological systems aspects and dynamics to be sufficiently responsive to COVID-19 type stresses in an integrated manner. These need to include locally specific systems at the landscape and jurisdictional level that will leverage socioecological system dynamics while also fulfilling the needs of communities directly or indirectly dependent on the agroecosystems. Transformative and reimaginative actions will be needed in the way humans related to nature for example. For instance, in the way humans interact with wildlife long-term to ensure that COVID-19 and Ebola type crises are minimized and better managed. Redesigning cities and also supply systems nationally and internationally to cater for basic food, equipment, and others during global "lockdowns" of the type seen during COVID-19 is also needed. These transformative and reimaginative actions would largely need to happen at macro and global levels and

largely across sectors given the nature of drivers. It would require rethinking development approaches and targets with an emphasis on sustainability going forward.

Overall, whether mitigative, adaptive, transformational, or reimaginative, all actions would need to be backed up by massive investments, policies, and incentives. Investments will have to be justified by meeting the expectations of the current and future generations. Above all, leadership, collaboration, and joint action will be needed if impacts from COVID-19 like stresses on socioecological systems would be minimized in the future.

**Supplementary Materials:** The following are available online at https://www.mdpi.com/2071-1050/13/3/1278/s1.

**Author Contributions:** Conceptualization, L.A.D., M.v.N. and P.A.M.; methodology, formal analysis, L.A.D., M.v.N., P.A.M. and K.M.; writing—original draft preparation, L.A.D.; writing—review and editing, L.A.D., M.v.N., P.A.M. and K.M. All authors have read and agreed to the published version of the manuscript.

**Funding:** The authors acknowledge the support of the Forest, Trees and Agroforestry Program of the CGIAR (FTA Flagship 5) for the financial support.

**Institutional Review Board Statement:** Not applicable.

**Informed Consent Statement:** Not applicable.

**Data Availability Statement:** The data used for the study is provided as Supplementary Material.

**Acknowledgments:** We are very grateful to Dibo Duba for her contributions on the literature data compilation. We also acknowledge the reviewers for the thoughtful and very useful inputs that helped improve the manuscript. The authors acknowledge the support of the Forest, Trees and Agroforestry Program (FTA Flagship 5) of the CGIAR for the financial support.

**Conflicts of Interest:** The authors declare no conflict of interest.

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
