# Peer review of "COVID-19 Pandemic and Agroecosystem Resilience: Early Insights for Building Better Futures"

_sustainability, doi:10.3390/su13031278_

Round 1
Reviewer 1 Report
I very much enjoyed reading this timely paper, particularly because last spring when we were forced to move toward online teaching only, it had a tremendous impact upon student learning in what is very much an applied field-based curriculum. We focused on how it has impacted individuals and personal resilance.
I very much enjoyed reading this timely paper, particularly because last spring when we were forced to move toward online teaching only, it had a tremendous impact upon student learning in what is very much an applied field-based curriculum in landscape ecology, planning and landscape architecture. I then attempted to focus on how the pandemic has impacted individual and personal resilance across cultures; this this manuscript was of great interest to me.
The abstract is very far reaching (in a positive way) and had me anticipating what would be reported in the manuscript, however I was concerned that to do so is perhaps not fully possible within the scope and size/scale of an academic paper but perhaps better suited to the formation of book chapters (hint-hint MDIP). If I find one disappointment in reading the manuscript, it is that much of the work and analysis herein focuses greatly upon agroecosystems and leaves me craving greater discussion upon resilience mechanisms social and stated ecological systems (as teased in the title of the paper and described in the abstract) without such a great emphasis upon agroecosystems. However, given the background of the authors, I will give it that concession.
The very conceptual schemes as presented in Figure 2A and perhaps also in 2B I found a bit confusing to understand, both in the graphics as presented and in the text descriptions. In 2A, there is no scale to the X and Y axis, making the presentation of carrying capacity, and factor such as disruptive elements, etc., very abstract – to me. How do I relate this? What might be the temporal scale? Is this applicable to developing world nations as well as EU and USA, for example? Given that these form the basis for how this paper wishes to present the schemes for dealing with, and providing guidance to, how society responds to the COVID pandemic and its broad reaching impacts, please spend more time explaining 2A and 2B. If this was introduced by the Resilience Alliance, then please provide a more robust introduction for the reader of this paper to these “conceptual” ideas. This was only later presented in Figure 5, however again it is not so clear what 5A and 5B are trying to say or how the reader relates this to so many broad schemes for building the reported better future. Perhaps if this were an oral presentation, this could be better conveyed to the audience or in this case, the reader.
What is retention / inclusion criteria mentioned on line 155 that formed the basis for your lit review? Perhaps the lit review could or should be published as an annotated bibliography?
Figure 3 had me curious as to why and how Wildlife would have a negative score of 58.8 and a positive score of 35.3, given our observed and field measured impacts of what Covid has done to the environment here in the EU, as in lessening observed impacts in less than one season or year. It was only until later in the paper with the discussion of impacts to forested areas and wildlife in rural areas, and protected landscapes in developing countries in Africa that it began to make sense. I must have missed the last sentence in 1.0 Introduction.
Fig 6 works to try and illustrate how wildlife conservation tries to respond to the associated impacts of Covid and movement of people from urban to rural locations (that in itself is a fascinating sociological study to be undertaken…). This and Fig 7 hints as what government could and should be doing to deal with the impacts to wildlife and protect area landscapes. What I miss is better presentation of how it can be combatted better in the real world. For example, if there has been a reverse migration from urban areas to rural landscapes (and family compounds traditional to parts of Africa), and if you are trying to limit impacts to agroforestry, agroenvironmental, and protected areas as people struggle to feed themselves and survive, rather than increased government enforcement or policing of protected areas, government should provide food and fuel resources to these rural locations rather than trying to fight opportunism of protected areas (sorry for the very long sentence). Empty stomachs do not make conservationists. My research in ecotourism documented this.
Otherwise I very much enjoyed this paper and it would be useful to also see how similar analysis of impacts to sociological / ecological systems in other economies is being dealt with, or not.
Author Response
Reviewer 1
Dear Reviewer,
We are so grateful for the very constructive comments you made on our manuscript. It has tremendously helped us to improve the manuscript. Find below the specific responses and highlights fo the changes we made based on your comments.
Comments and Suggestions for Authors
I very much enjoyed reading this timely paper, particularly because last spring when we were forced to move toward online teaching only, it had a tremendous impact upon student learning in what is very much an applied field-based curriculum. We focused on how it has impacted individuals and personal resilance.
Response: Thank you for your feedback. We are very pleased to read that you have enjoyed reading the manuscript.
I very much enjoyed reading this timely paper, particularly because last spring when we were forced to move toward online teaching only, it had a tremendous impact upon student learning in what is very much an applied field-based curriculum in landscape ecology, planning and landscape architecture. I then attempted to focus on how the pandemic has impacted individual and personal resilance across cultures; this this manuscript was of great interest to me.
Response: Thank you for your feedback.
The abstract is very far reaching (in a positive way) and had me anticipating what would be reported in the manuscript, however I was concerned that to do so is perhaps not fully possible within the scope and size/scale of an academic paper but perhaps better suited to the formation of book chapters (hint-hint MDIP). If I find one disappointment in reading the manuscript, it is that much of the work and analysis herein focuses greatly upon agroecosystems and leaves me craving greater discussion upon resilience mechanisms social and stated ecological systems (as teased in the title of the paper and described in the abstract) without such a great emphasis upon agroecosystems. However, given the background of the authors, I will give it that concession.
Response: Thank you. We have revised the title of the manuscript so that it captures the main emphasis of the manuscript that is the agroecosystems. Here the agroecosystem seen as a complex subset of social-ecological systems. See revised title [COVID-19 pandemic and agroecosystem resilience: Early in-sights for building better futures].
We also noted your comment on the abstract and we have revised it to focus more on the content of the manuscript with less of beyond the manuscript conceptual projections. See the revised abstract.
The very conceptual schemes as presented in Figure 2A and perhaps also in 2B I found a bit confusing to understand, both in the graphics as presented and in the text descriptions. In 2A, there is no scale to the X and Y axis, making the presentation of carrying capacity, and factor such as disruptive elements, etc., very abstract – to me. How do I relate this? What might be the temporal scale? Is this applicable to developing world nations as well as EU and USA, for example? Given that these form the basis for how this paper wishes to present the schemes for dealing with, and providing guidance to, how society responds to the COVID pandemic and its broad reaching impacts, please spend more time explaining 2A and 2B. If this was introduced by the Resilience Alliance, then please provide a more robust introduction for the reader of this paper to these “conceptual” ideas. This was only later presented in Figure 5, however again it is not so clear what 5A and 5B are trying to say or how the reader relates this to so many broad schemes for building the reported better future. Perhaps if this were an oral presentation, this could be better conveyed to the audience or in this case, the reader.
Response: Thank you. Yes, the panarchy concept introduced by the Resilience Alliance that we used in the manuscript might seem a little complicated. However, for the sake of the readers we have introduced some sentences that elaborate the process. These details are also added for the DPSIR (Driver, Pressure, State, Impact, Response) framework too. See second paragraph of section 2.2. We acknowledge that it may not be straightforward to grasp the complex nature of interactions and interdependencies in agroecosystems and social-ecological systems. To help even on this, we have redesigned the frameworks to shown the directional movement of processes to guide readers. See Figure 2a, 4a and 4b.
What is retention / inclusion criteria mentioned on line 155 that formed the basis for your lit review? Perhaps the lit review could or should be published as an annotated bibliography?
Response: we have revised section 2.3 to make explicit the criteria used for retention or elimination. See paragraph 1-3 of section 2.3.
Figure 3 had me curious as to why and how Wildlife would have a negative score of 58.8 and a positive score of 35.3, given our observed and field measured impacts of what Covid has done to the environment here in the EU, as in lessening observed impacts in less than one season or year. It was only until later in the paper with the discussion of impacts to forested areas and wildlife in rural areas, and protected landscapes in developing countries in Africa that it began to make sense. I must have missed the last sentence in 1.0 Introduction.
Response: Yes, the emphasis is on developing countries.
Fig 6 works to try and illustrate how wildlife conservation tries to respond to the associated impacts of Covid and movement of people from urban to rural locations (that in itself is a fascinating sociological study to be undertaken…). This and Fig 7 hints as what government could and should be doing to deal with the impacts to wildlife and protect area landscapes. What I miss is better presentation of how it can be combatted better in the real world. For example, if there has been a reverse migration from urban areas to rural landscapes (and family compounds traditional to parts of Africa), and if you are trying to limit impacts to agroforestry, agroenvironmental, and protected areas as people struggle to feed themselves and survive, rather than increased government enforcement or policing of protected areas, government should provide food and fuel resources to these rural locations rather than trying to fight opportunism of protected areas (sorry for the very long sentence). Empty stomachs do not make conservationists. My research in ecotourism documented this.
Response: WE fully agree with your statement on this matter. The conclusion section in fact delves very much in emphasizing the importance of holistic approaches and the need for emphasizing short, medium and long term needs of the ecosystem and the communities if the ‘better futures’ context is to be effective. We have also added sentences that highlight the needs of the local people if any future engagement for sustainability need to be feasible and practical. See paragraph 3 of the conclusion section.
Otherwise I very much enjoyed this paper and it would be useful to also see how similar analysis of impacts to sociological / ecological systems in other economies is being dealt with, or not.
Response: we greatly appreciate your comments and we are monitoring the emerging evidences and workable frameworks that could help us do the analysis at a larger social-ecological scale and we would be very pleased to partner on that.

Reviewer 2 Report
It is better to provide an example of the social-ecological system you will be dealing with in this manuscript at earliest so that the readers have some ideas about what to expect.
It is better to write the “adaptive cycle” rather than “The r-K-Ω-α cycle….”
Figure 2A should look more like a “horizontal 8”. Please refer to Holling’s papers or books.
I think the conceptual frameworks (panarchy and pressure-state-response) should be elaborated a bit.
What is the reason for the restriction to Web of Science for literature search? Please mention in the text.
Figure 3 and Figure 4 essentially represent the same information. I would suggest putting one in the appendix.
Line 205: Should be “COVID-19”.
Table 4: “R&D” or “R4D”?
Line 478: “such as”?
In the methods section, it is mentioned that an Excel sheet of categories is appended. But in the “Supplementary Materials” of the manuscript part, the response is “No”. Please fix it.
Author Response
Reviewer 2
Comments and Suggestions for Authors
Dear Reviewer,
We are so grateful for the very constructive comments you made on our manuscript. It has tremendously helped us to improve the manuscript. Find below the specific responses and highlights fo the changes we made based on your comments.
It is better to provide an example of the social-ecological system you will be dealing with in this manuscript at earliest so that the readers have some ideas about what to expect.
Response: the emphasis of the manuscript is now on agroecosystems which are largely understandable by wider audience. This change is effected from the title all the way through to the body of the manuscript. The changes do not induce any conceptual deviation because agroecosystems are complex subsets of the social-ecological systems. We have introduced it from the first sentence of the Introduction section.
It is better to write the “adaptive cycle” rather than “The r-K-Ω-α cycle….”
Response: Thank you. We have made the suggested changes. See line line 118 of the revised document.
Figure 2A should look more like a “horizontal 8”. Please refer to Holling’s papers or books.
Response: Thank you. See the revised figure 2a, 4a, 4b. The revised figure is taking that horizontal shape captured well.
I think the conceptual frameworks (panarchy and pressure-state-response) should be elaborated a bit.
Response: Thank you. Yes, the panarchy concept introduced by the Resilience Alliance that we used in the manuscript might seem a little complicated. However, for the sake of the readers we have introduced some sentences that elaborate the process. These details are also added for the DPSIR (Driver, Pressure, State, Impact, Response) framework too. See second paragraph of section 2.2. We acknowledge that it may not be straightforward to grasp the complex nature of interactions and interdependencies in agroecosystems and social-ecological systems. To help even on this, we have redesigned the frameworks to shown the directional movement of processes to guide readers. See Figure 2a, 4a and 4b.
What is the reason for the restriction to Web of Science for literature search? Please mention in the text.
Response: Thank you. We have clarified this the methods section specifically in line 161-175. The details provided have clarified the criteria.
Figure 3 and Figure 4 essentially represent the same information. I would suggest putting one in the appendix.
Response: Thank you. We have removed the old figure 4 which more or less presents a similar detail.
Line 205: Should be “COVID-19”.
Response: We have fixed this.
Table 4: “R&D” or “R4D”?
Response: We have fixed this.
Line 478: “such as”?
Response: We have fixed this.
In the methods section, it is mentioned that an Excel sheet of categories is appended. But in the “Supplementary Materials” of the manuscript part, the response is “No”. Please fix it.
Response: We have provided the file for this.
